# Malignant Pleural Effusions Impact on Fatigue (IMPE-F): A Prospective Observational Cohort Pilot Study

**Avinash Aujayeb** [1,*] and **Donna Wakefield** [2]

1   Respiratory Department, Northumbria Healthcare NHS Foundation Trust, Northumbria Way, Cramlington NE23 6NZ, UK
2   Specialist Palliative Care Team, North Tees & Hartlepool NHS Foundation Trust, Farndale House, University Hospital of North Tees, Hardwick Rd, Hardwick, Stockton-on-Tees TS19 8PE, UK; donna.wakefield1@nhs.net
*   Correspondence: avinash.aujayeb@nhct.nhs.uk; Tel.: +44-7703343329

**Abstract:** Introduction: Cancer-related fatigue is well described. Fatigue in patients with a malignant pleural effusion (MPE) has not been directly studied. Methods: A prospective observational cohort pilot study 'Do Interventions for Malignant Pleural Effusions (MPE) impact on patient reported fatigue levels (IMPE-F study)' is planned to determine whether pleural interventions reduce fatigue in MPE. Fatigue will be assessed with a validated patient reported outcome measure, FACIT-F. Discussion: MPE-F has funding from Rocket Medical Plc, and is part of a Masters in Clinical Research at Newcastle University. Respondent fatigue will be addressed by the investigators going through the questionnaire with the participants. Inclusion criteria are all patients above 18 years of age with a presumed MPE undergoing a procedure and able to consent. The expected number of participants is 50. Trial registration: The IMPE-F study has Research Ethics Committee (REC) [20/YH/0224] and Health Research Authority (HRA) and Health and Care Research Wales (HCRW) approvals [IRAS project ID: 276451]. The study has been adopted on National Institute for Health Research portfolio [CPMS ID 46430].

**Keywords:** malignant pleural effusion; fatigue; cancer





## 1. Introduction

Malignant pleural effusions (MPE) are a common presentation with around 40,000 new cases in the UK each year [1]. The incidence is rising [1]. The development or presence of MPE confers a poor prognosis, with median survival of 3–12 months depending on the type of the primary cancer and the performance status [2]. MPE is associated with significant symptom burden as well as often requiring more than one pleural intervention, hence consuming significant resources [3].

Pleural interventions can be regular therapeutic aspirations or thoracenteses, intercostal drain insertion with talc pleurodesis, indwelling pleural catheter insertion with or without talc instillation or medical (local anesthetic) or video assisted thoracoscopy [3]. A patient-centered approach where patients can choose between the various modalities is advocated and patient education plays an important role in this approach [4].

The quality of life of patients with MPE has been studied before and recently summarized in a systematic review [5]. Sivakumar et al., found that pleural interventions improve health related quality of life at 4–12 weeks but the studies had high attrition rates. Most of the reviewed studies calculate outcomes from surrogate endpoints instead of using validated quality of life tools. Breathlessness is also the most commonly studied symptom and it is noted that research in overall wellbeing, social and functional factors is lacking. Anecdotally, pleural interventions are offered to patients to relieve breathlessness specifically.

Cancer related fatigue is well described. Cancer-related fatigue is a subjective symptom experienced by patients at all stages of disease and can occur during treatment,

in advanced disease and in disease free survivors [6–8]. Approximately 40% of cancer patients have experienced fatigue at the time of diagnosis and up to 90% experience this symptom during anti-cancer treatment such as radiotherapy or chemotherapy [6–8].

The available literature on PubMed was reviewed. A search with the keywords ((malignant pleural effusion) AND (chest drain OR pleural aspiration OR indwelling pleural catheter)) AND (quality of life OR fatigue) yielded 130 results but none were related to fatigue and were mostly related to dyspnoea. A second search with the keywords ((malignant pleural effusion) AND (chest drain OR pleural aspiration OR indwelling pleural catheter)) AND (fatigue) yielded 4 irrelevant results. A third search (malignant pleural effusion) AND (fatigue) yielded 45 results and two studies were further analyzed. Fatigue is a recognized symptom in patients with MPE but has only been studied as part of more global quality of life studies [9,10].

Sabur et al., measured the impact of tunneled pleural catheters on broad measures of quality of life in 83 patients with MPE [9]. They used two questionnaires, the generic QLQ-C30 oncology trial quality of life questionnaire and the LC13 supplemental questionnaire for lung cancer, which measures symptoms associated with lung cancer and treatment side effects. The former includes a component on fatigue but that is not as specific. Significant improvement in fatigue was noted 2 weeks after treatment, but this was not sustained at 14 weeks.

Dresler et al., compared thoracoscopy with talc insufflation to thoracostomy and talc slurry for 482 patients with MPE [10]. The primary outcome was non-recurrence of radiographic MPE; however, symptoms were also measured using the QLQ-C30. The two treatments had similar impacts on quality of life, however the talc insufflation arm demonstrated a decrease in fatigue, compared to increased fatigue for patients who received talc slurry.

We hypothesized that high levels of fatigue are prevalent in patients with MPE and that the assessment of fatigue using the validated FACIT-Fatigue scale [11] is feasible, thus forming the basis for further studies on fatigue in MPE using the FACIT-F scale.

From November 2019 to March 2020, patients with MPE or presumed MPE presenting to the regional pleural service were surveyed. Consent was verbally obtained and local governance approval was sought (Reference C3421). Basic demographics, associated co-morbidities and relevant hematological results were collected. Patients self-reported fatigue levels by completing the FACIT-F tool by themselves. Descriptive statistical methodology was applied. Clinical care proceeded as normal.

30 patients were surveyed (12 female and 18 male). Median age was 74.8 years, IQR 16, range (46–87). Diagnoses were 15 pleural mesotheliomas, 9 lung carcinomas, 4 breast cancers and 2 ovarian cancers. In total, 14 of the mesothelioma patients were male. Patients had a wide range of co-morbidities: hypertension (n = 18), previous myocardial infarctions (n = 5), diabetes (n = 4), hyperthyroidism (n = 1), chronic obstructive pulmonary disease (n = 2) and previous resected cancers (n = 7). Three of the previous resected cancers were breast carcinomas which had now metastasized. All hemoglobin and renal function were within the normal range. ECOG performance status was between 1 and 3 (1: n = 16, 2: n = 12, 3: n = 2). None of the patients were on opioid drugs or suffered from anxiety or depression. There are a number of limitations to the above data. We did not correlate the score to time since diagnosis. We did not correct for whether the patient was on chemotherapy or had recurrent effusions or assessed if any had undergone pleural interventions. There was no control group, the scores were not evaluated over time.

The primary aim of this small local survey was to assess the incidence of fatigue in patients presenting with MPE. The FACIT-Fatigue scale was used as it has been validated for assessing cancer-related fatigue [11]. It is a 13-question survey where patients self-rate their fatigue level over the previous 7 days. A previous systematic review identified FACT F (previously called the Functional Assessment of Cancer Therapy-Fatigue (FACT-F)] as one of the two most commonly used scales to assess fatigue, the other being is the European Organization for Research and Treatment of Cancer Quality of Life Questionnaire (EORTC

QLQ C30) (fatigue subscale). The inclusion criteria of the review methodology required were robust. The scales needed to have been validated in cancer patients and widely used. Scales also had to have a minimum quality score. FACT F met those criteria and was found to be user friendly. In total, 25 out of 30 participants (83%) answered yes to 'I feel fatigued' with 27 out of 30 (90%) saying they felt weak all over. There were 26 respondents for the last two questions which demonstrated high levels of frustration at the participants not being able to do the things they wanted to do and limitations of social activities. Participants were willing to complete the 13-question survey by themselves in clinic, without supervision. Not all points in the survey were answered, perhaps due to respondent fatigue which is a well-documented phenomenon where survey participants become tired of the task and the quality of the data provided then begins to deteriorate as attention and motivation drop toward later sections of a questionnaire.

The lack of direct evidence of fatigue in malignant pleural effusions, whether any pleural interventions provide sustained improvement in fatigue and whether any confounding factors should be accounted for led us to the setting up of the Interventions for Malignant Pleural Effusions impact on Fatigue (IMPE-F) study.

*Aims and Objectives*

The IMPE-F study will explore if pleural interventions to treat malignant pleural effusions have an impact on patient reported levels of fatigue. Specific co-founders will be accounted for. This is discussed further in the outcomes section.

## 2. Experimental Section

### 2.1. Study Design

The IMPE-F study is a prospective observational cohort pilot study and will be done in conjunction with routine clinical care. It is being performed as part of a Masters in Clinical Research at Newcastle University (Annex 1 and 2). Research Ethical Committee: 20/YH/0224 and date of approval: 31 July 2020.

### 2.2. Setting and Centre

Northumbria Healthcare NHS Foundation Trust has a well-established pleural service, led by a full-time pleural consultant and pleural fellow and supported by 2 other respiratory consultants [12]. Pleural clinics were held weekly with fortnightly theatre lists where procedures performed include local anesthetic thoracoscopy (LAT), indwelling pleural catheter (IPC) insertion and removal. The trust has a flagship acute care centre with 2 respiratory inpatient wards which provide inpatient services for chest drain insertion, talc pleurodesis and IPC insertion if required.

### 2.3. Trial Population

Inclusion criteria (Annex 3–5) are

- Age 18–99;
- Any sex;
- Any ethnicity;
- Presence of a or presumed malignant pleural effusion (MPE);
- Attending the Northumbria pleural service to have an intervention to treat MPE;
- Having sufficient understanding of the English language to give informed consent;
- Having capacity to make decision to take part.

Exclusion criteria are being under 18 years, having a non-malignant pleural effusion, not having treatment for MPE, lacking capacity to give informed consent and not understanding sufficient English to give informed consent.

### 3. Patient Approach, Consent and Sampling

*3.1. Size of Sample*

This is a pilot to test the feasibility of the study and so a statistical power calculation is not required. Over 300 procedures are done yearly locally [12] and thus a recruitment target of 50 seems reasonable.

*3.2. Sampling Technique*

Every appropriate patient will be offered the opportunity to take part and details entered into the screening log. No randomization is required.

*3.3. Sample Identification*

Participants will be identified by the usual clinical care team when they are seen in clinic (or as inpatients). Only after the patient has given written informed consent will the researcher be able to access the patient records. Patients will not be recruited by any other method such as posters, leaflets or websites. Participants will be recruited when they attend for usual clinical care. No additional visits are necessary. Participants will consent to receive follow up phone calls from the research team (phone call will take around 5 min). There will be no cost to the participants. No costs will be paid for participant's time and this is made clear in the patient information sheet (PIS, Supplementary Materials) and will be re-iterated during the consent process. A limitation of the above survey of 30 patients was clinic recruitment and they had a higher performance status than encountered in the inpatient service where patients normally present with advanced metastatic disease and poorer performance status. Patients with advanced disease welcome the opportunity to take part in research but are less likely to have the opportunity [13–15]. Thus, we will aim to recruit patients with advanced disease, even when prognosis is short.

*3.4. Consent*

Gaining informed consent will involve a discussion between the potential participant and an individual knowledgeable about the research, about the nature and objectives of the study and possible risks associated with their participation. There will be ample time to ask questions.

*3.5. Study Interventions*

This is a low burden study with no change to usual patient care. Completing the FACIT-F questionnaire prior to a procedure which takes less than 5 min and then a phone call to repeat the same questions at 7, 14 and 30 days. The main burden is the time and inconvenience of completing a questionnaire, but we believe this to be minimal. Respondent fatigue will be addressed by the investigators going through the questionnaire with the participants.

The attrition rate of MPE is high [1–3,5] and it is very likely that participants may die during the follow up period. A previous systematic review of QoL after interventions for MPEs found a short follow up period [5]. Every effort will be made to check using the electronic patient records that patients have not been admitted to hospital or have died to avoid calling and cause family distress. One of the researchers is a palliative medicine consultant with advanced communication skills training and experience of discussing sensitive issues with patients and bereaved relatives. Doing research in patients with palliative needs comes with the risk of death during follow up and potential carer distress. However, it is important that this patient group do not miss out on the opportunity to take part in research because of fear of causing upset. Sensitive communication will help minimize any potential distress.

*3.6. Primary and Secondary Outcomes*

The primary outcome is to determine if drainage of malignant or presumed malignant pleural effusions has an impact on fatigue, as measured by the FACIT-fatigue scale. Minimal

score is 0 and maximal score is 52, higher scores suggesting increasing fatigue). Score will be done on the day of the drainage, before any procedure, then repeated at 7, 14 and 30 days. One of the secondary outcomes is to determine if the type of procedure and amount of fluid drained has any effect on fatigue. The other secondary outcome is to determine if the above methods are suitable for a larger statistically powered study, i.e., are participants willing to take part or not.

*3.7. Data Collection*

Data will be collected onto purpose-designed case report forms (CRFs) and participant-completed questionnaires and entered onto a bespoke database for data cleaning and analysis. Access to the database will be via a secure password-protected National Health Service (NHS) server. Patient identifiable information (name and telephone number) will need to be initially stored to allow follow up at 7, 14 and 30 days. At the end of the data collection period, each patient will be allocated a serial number. Anonymized patient data will then be compiled for statistical analysis. The following data will be also be gathered onto the CRFs: sex, diagnosis (site/type of cancer and whether metastatic), comorbidities (esp. lung disease), time since cancer diagnosis, time since last anti-cancer therapy/if currently having chemotherapy or radiotherapy, Integrated Patient Outcome Score (I POS) to assess symptom burden & screen for depression, size of pleural effusion & if unilateral/bilateral, amount of fluid drained, performance status, hemoglobin and creatinine levels.

*3.8. Risk of Bias*

This survey-based study is subject to response bias, which is a generic term for the tendency of participants to respond inaccurately or falsely to questions. A form of response bias, acquiescence bias, also known as "yea-saying", is where participants tend to agree with all the questions. We have a balanced response set with a number of positively and negatively worded questions. Demand characteristics is another type of response bias where participants alter responses or behaviors because of active engagement in the study and thus adopt behaviors they believe belong in a study. We will reduce demand characteristics is by being as neutral as possible.

*3.9. Patient and Public Involvement*

Patients have been asked to comment on the acceptability of the research, to trial the questionnaire for acceptability and give feedback on Participant Information Sheets. A group will also be invited to review the results and discuss dissemination of study findings.

**4. Statistical Analyses**

Analysis will follow Consolidated Standards of Reporting Trials (CONSORT) guidelines. Analyses will be conducted with appropriate statistical software. Multiple imputation methods will be used if greater than 5% of cases have missing data, otherwise complete case analysis will be undertaken. Compliance rates will be reported, including the number of participants who have withdrawn from the study, have been lost to follow-up or died. Subgroup analyses according to pleural intervention or cancer types are planned. No interim analyses are planned. The primary analysis will take place when follow-up is complete for all recruited participants.

**5. Ethics and Dissemination**

The pleural interventions are already routinely used in the NHS. The pilot study is managed by Northumbria HealthCare Research and Development and sponsored by Northumbria HealthCare NHS Foundation Trust. Rocket Medical Plc have contributed 250 £ to the costs of the study. Each participant has the right to withdraw at any time. The investigator may withdraw the participant if a clinical reason is discovered, for example

that the effusion is not a malignant one. If a participant wishes to withdraw, any data already collected will be included in the study analyses, unless the participant expresses a wish for their data to be excluded. The findings will be disseminated at international meetings and in peer-reviewed publications.

## 6. Study Progress

Recruitment started in November 2020.

## 7. Limitations and Future Directions

It is difficult to differentiate between fatigue due to a cancer and the related malignant effusion, as an effusion often signifies disseminated disease. Prue et al., describe attempts at regression analysis to adjust for confounding factors, attempts at recruitment from only one stage of illness or at the same from treatment and found that the number of variables made this incredibly complex [16]. Minton et al., concluded that treatment history and hemoglobin levels had the strongest association with severe fatigue levels [17]. A regression model could be developed with their large number of patients. Within our pilot study, in depth statistical analysis will not be possible with a smaller number of patients. However, confounding factors will be collected to assess any trends. We acknowledge that the primary cause of fatigue might not be easily identifiable. If our methodology is acceptable then a larger adequately powered study will be commenced. Non-malignant effusions can also cause fatigue, and perhaps further studies could include this group as a control, with blinding to those performing statistical analysis.

**Supplementary Materials:** The following are available online at https://www.mdpi.com/article/10.3390/jor1020011/s1, Ethical and Health Research Authority decision letters (Annex 1 and 2), patient information sheet (Annex 3), source document (Annex 4) and patient consent (Annex 5).

**Author Contributions:** Conceptualization, A.A. and D.W.; methodology, A.A. and D.W.; writing—original draft preparation, A.A. and D.W.; writing—review and editing, A.A. and D.W. All authors have read and agreed to the published version of the manuscript.

**Funding:** This research was funded by Rocket Medical Plc, for 250 £ for administrative costs. No Other funding is available.

**Institutional Review Board Statement:** Research Ethical Committee reference 20/YH/0224 dated 31 July 2020.

**Informed Consent Statement:** Informed consent was obtained from all subjects involved in the study.

**Data Availability Statement:** Raw data is available upon request.

**Acknowledgments:** We thank Jane Luke, Jemma Fenwick, Lorelle Dismore and Diane Charlton for administrative and technical support.

**Conflicts of Interest:** The authors declare no conflict of interest. The funders had no role in the design of the study; in the collection, analyses, or interpretation of data; in the writing of the manuscript, or in the decision to publish the results.

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
