# Peer review of "Malignant Pleural Effusions Impact on Fatigue (IMPE-F): A Prospective Observational Cohort Pilot Study"

_2673-527X, doi:10.3390/jor1020011_

Round 1

Reviewer 1 Report

the idea of studying the dynamics of fatigue as a result of various therapeutic interventions in patients with malignant pleural effusions is excellent.

the draft is generally well done apart from the part regarding protocol description.

instead of being generic please be specific on the following issues:

1 outcome measures (ie fatigue is going to be measured with X scale)

2 duration of observation:  when is the baseline (time of diagnosis or time of intervention?) ;  when is the next time point(s) for observation compared to baseline

3 primary and secondary endpoints

Author Response

the idea of studying the dynamics of fatigue as a result of various therapeutic interventions in patients with malignant pleural effusions is excellent. Thank you

the draft is generally well done apart from the part regarding protocol description. thank you

instead of being generic please be specific on the following issues:

1 outcome measures (ie fatigue is going to be measured with X scale) I agree with this. so have been more specific by saying we will measure fatigue using the facit f questionnaire and quantifying the scale.

2 duration of observation:  when is the baseline (time of diagnosis or time of intervention?) ;  when is the next time point(s) for observation compared to baseline : I have mentioned this : day of procedure (pre-procedure), 7, 14 and 30 days

3 primary and secondary endpoints 

Secondary endpoints included considering whether the different procedures may have an impact on the outcome. Since this is a pilot study- we are analysing whether this methodology is feasible for a larger study or are there any aspects of the methodology which need adjustment in order to ensure a successful larger study- one important aspect of this being, are patients willing to take part or are they too fatigued? I have been more specific about this in the text and re-arranged some of the data we are collecting into another paragraph

Reviewer 2 Report

Dear Editor,

Dear Authors,

I found the manuscript entitled “Malignant Pleural Effusions Impact on Fatigue 3 (IMPE-F): A Prospective Observational Cohort Pilot 4 Study” presenting protocol of the study, well planned and written. It deals with important problem of fatigue in patients with malignant pleural effusion. Although problem of quality of life is frequently studied in different groups of patients undergoing various procedures, the fatigue remains still not well understood issue. I have only doubt, if the problem influenced by so many different factors (e.g. comorbidities, age, performance status, the degree of nutrition etc), is possible to be appropriately assessed in regard to pleural procedures in a small, diverse group of patients. But maybe this pilot study will give the answer and indications for further research.

I have only some minor comments/corrections:

-page1, line34 – at the end of the sentence, after “thoracoscopy” there is “3”, not marked as reference. Is that correct?

- line 84 - I would recommend to replace “3” at the beginning of the sentence with “Three”

-In inclusion criteria consider to remove “must” or replace it with another word.

-141 –I do not understand what does “12” relate to in the sentence:” Over 300 procedures are done yearly locally, 12 and thus a recruitment target of 50 seems reasonable.” Is the sentence correct?

-165-166 - what questionnaire do you mean – is it FACIT-F or another self-prepared questionnaire?

-line 188- there is an additional dot at the end of the sentence – to be deleted.

I wish you good luck with publication process and during the study.

Author Response

-page1, line34 – at the end of the sentence, after “thoracoscopy” there is “3”, not marked as reference. Is that correct? This was meant to be a reference and so it has been corrected

- line 84 - I would recommend to replace “3” at the beginning of the sentence with “Three” This has been corrected

-In inclusion criteria consider to remove “must” or replace it with another word. I have removed all the ‘must’ and replaced them appropriately

-141 –I do not understand what does “12” relate to in the sentence:” Over 300 procedures are done yearly locally, 12 and thus a recruitment target of 50 seems reasonable.” Is the sentence correct? The number 12 is meant to be a reference and I have annotated it as such

-165-166 - what questionnaire do you mean – is it FACIT-F or another self-prepared questionnaire? I have made this more specific and yes this is the FACITF questionnaire

-line 188- there is an additional dot at the end of the sentence – to be deleted. I have deleted this

I wish you good luck with publication process and during the study. Thank you.

Reviewer 3 Report

The manuscript by Avinash Aujayeb and Donna Wakefield entitled “Malignant Pleural Effusions Impact on Fatigue (IMPE-F): A Prospective Observational Cohort Pilot Study” reports a protocol to study the correlation between fatigue and malignant pleural effusions.

Both malignant pleural effusion and fatigue are symptoms of cancers or tumors. Malignant pleural effusions mostly occur in late stage malignant patients. Fatigue emerges in most of the malignant patients. How to differentiate the fatigue from cancer itself or from pleural effusion?

Does non-malignant pleural effusion cause fatigue and that would be a control for malignant pleural effusion caused fatigue?  

It is crucial to define what is a fatigue and the severity or degree of the fatigue in the text. Please bring the design and questionnaire to the surface but not as a supplementary source file.

Though it is a protocol, it is important to include data in the pilot study.

Author Response

How to differentiate the fatigue from cancer itself or from pleural effusion? 

thank you for this: this is a long response, but i have elaborated on this in a new section called 'limitations and future directions'

There are many factors associated with fatigue, which makes studying one potential cause of fatigue complex. A systematic review (Prue G, Rankin J, Allen J, Gracey J, Cramp F. Cancer-related fatigue: a critical appraisal. European Journal of Cancer. 2006 May 1;42(7):846-63.) highlighted that a number of previous studies undertook regression analysis in an attempt to adjust for confounding factors, whilst others aimed to recruit participants from a uniform stage of illness or at the same standardised time from treatment, however the wide breadth of variables made this incredibly complex. Further studies aimed to investigate which factors are independently most associated with cancer-related fatigue (Minton O, Strasser F, Radbruch L, Stone P. Identification of factors associated with fatigue in advanced cancer: a subset analysis of the European palliative care research collaborative computerized symptom assessment data set. Journal of pain and symptom management. 2012 Feb 1;43(2):226-35.), this concluded that treatment history and haemoglobin levels had the strongest association with severe fatigue levels. This study had a large number of patents (n=720) and so a logistic regression model could be developed. Within our pilot study, only a small number of participants will be involved (n=30) and so meaningful statistical analysis will not be possible. However, it is helpful to gather data on confounding factors to assess if those patients with the confounding factors most strongly associated with fatigue (such as ongoing chemotherapy or low haemoglobin levels) or those with multiple potential cofounding factors are more likely to have persistent fatigue than those without such factors present. We acknowledge as a limitation that we do not know if the primary cause of fatigue in these patients is related to the pleural effusion, the cancer itself or other confounding factors,  however our aim is to identify if a possible intervention can have an impact and improve fatigue levels in  this patient group, regardless of the primary cause. If the methodology of this pilot study is found to be acceptable then a larger study will be commenced, where statistical analysis can be performed to investigate on a larger scale the hypothesis that treatment of pleural effusions can have an impact on fatigue.

Does non-malignant pleural effusion cause fatigue and that would be a control for malignant pleural effusion caused fatigue?  

Non-malignant pleural effusions may cause fatigue. As described previously, there are many confounding factors for fatigue that we are unable to control for. Fatigue in patients with cancer is much higher than in the general population, the mechanism for this is unclear. By limiting our study to include only patients with a cancer diagnosis it is helpful to ensure our population potential have fatigue for similar reasons. It is also important as the presence of a malignant pleural effusion have a poor prognosis and so it is vital to make treatment decisions rapidly,   to optimise symptom control early and before end of life. After the pilot study, a large study is planned, this could potentially include a control group of those with non-malignant effusions, with blinding to those performing statistical analysis. 

It is crucial to define what is a fatigue and the severity or degree of the fatigue in the text. Please bring the design and questionnaire to the surface but not as a supplementary source file. Though it is a protocol, it is important to include data in the pilot study.

We have already defined fatigue as per Reviewer 1's comments, as well as the minimum and maximum. I am happy if the Editor in Chief wants to bring the FACIT F questionnaire into the front of the manuscript rather than a source file. I don't think it adds much but happy either way.